 eLife

# Fundamental constraints in synchronous muscle limit superfast motor control in vertebrates

Andrew F Mead[1], Nerea Osinalde[2†‡], Niels Ørtenblad[3†], Joachim Nielsen[3†], Jonathan Brewer[2†], Michiel Vellema[4†], Iris Adam[5], Constance Scharff[5], Yafeng Song[6], Ulrik Frandsen[3], Blagoy Blagoev[2], Irina Kratchmarova[2], Coen PH Elemans[4]*

[1]Department of Biology, University of Vermont, Burlington, United States; [2]Department of Biochemistry and Molecular Biology, University of Southern Denmark, Odense, Denmark; [3]Department of Sports Science and Clinical Biomechanics, University of Southern Denmark, Odense, Denmark; [4]Department of Biology, University of Southern Denmark, Odense, Denmark; [5]Institute of Biology, Freie Universität Berlin, Berlin, Germany; [6]Perelman School of Medicine, University of Pennsylvania, Philadelphia, United States

*For correspondence:
coen@biology.sdu.dk

[†]These authors contributed equally to this work

Present address: [‡]Department of Biochemistry and Molecular Biology, University of the Basque Country UPV/EHU, Vitoria-Gasteiz, Spain

Competing interests: The authors declare that no competing interests exist.

**Abstract** Superfast muscles (SFMs) are extremely fast synchronous muscles capable of contraction rates up to 250 Hz, enabling precise motor execution at the millisecond time scale. SFM phenotypes have been discovered in most major vertebrate lineages, but it remains unknown whether all SFMs share excitation-contraction coupling pathway adaptations for speed, and if SFMs arose once, or from independent evolutionary events. Here, we demonstrate that to achieve rapid actomyosin crossbridge kinetics bat and songbird SFM express myosin heavy chain genes that are evolutionarily and ontologically distinct. Furthermore, we show that all known SFMs share multiple functional adaptations that minimize excitation-contraction coupling transduction times. Our results suggest that SFM evolved independently in sound-producing organs in ray-finned fish, birds, and mammals, and that SFM phenotypes operate at a maximum operational speed set by fundamental constraints in synchronous muscle. Consequentially, these constraints set a fundamental limit to the maximum speed of fine motor control.

DOI: https://doi.org/10.7554/eLife.29425.001

## Introduction

Superfast muscles (SFMs) are the fastest known synchronous muscle phenotypes in vertebrates (*Rome, 2006*). Their ability to repetitively contract and relax fast enough to produce work at cycling rates over around 90 Hz and up to 250 Hz sets them apart from other muscles by almost two orders of magnitude and allows the execution of central motor commands with millisecond temporal precision (*Rome, 2006*; *Rome et al., 1988*; *Elemans et al., 2004*, *2008*, *2011*). The phenotype is defined by its mechanical performance and therefore establishing a muscle to be superfast requires quantification of its performance-profile (*Rome, 2006*). Although once considered an extremely rare muscle phenotype, SFMs have now been established in most major vertebrate lineages (mammals [*Elemans et al., 2011*], birds [*Elemans et al., 2004*; *Elemans et al., 2008*], reptiles [*Rome et al., 1996*], ray-finned fish [*Rome et al., 1996*]). SFMs in the bat larynx control the rapid, high frequency calls used by laryngeally echolocating bats to detect, orient to and track prey (*Elemans et al., 2011*). SFMs in the songbird syrinx control the precisely timed, rapidly produced acoustic elements and frequency sweeps of bird vocalizations (*Elemans et al., 2004*, *2008*) used for territorial defense

**eLife digest** Across animals, different muscle types have evolved to perform vastly different tasks at different speeds. For example, tortoise leg muscles move slowly over several seconds, while the flight muscles of a hummingbird move quickly dozens of times per second. The speed record holders among vertebrates are the so-called superfast muscles, which can move up to 250 times per second. Superfast muscles power the alarming rattle of rattlesnakes, courtship calls in fish, rapid echolocation calls in bats and the elaborate vocal gymnastics of songbirds. Thus these extreme muscles are all around us and are always involved in sound production.

Did superfast muscles evolve from a common ancestor? And how do different superfast muscles achieve their extreme behavior? To answer these questions, Mead et al. studied the systems known to limit contraction speed in all currently known superfast muscles found in rattlesnakes, toadfish, bats and songbirds. This revealed that all the muscles share certain specific adaptations that allow superfast contractions. Furthermore, the three fastest examples – toadfish, songbird and bat – have nearly identical maximum speeds. Although this appears to support the idea that the adaptations all evolved from a shared ancestor, Mead et al. found evidence that suggests otherwise. Each of the three superfast muscles are powered by a different motor protein, which argues strongly in favor of the muscles evolving independently. The existence of such similar mechanisms and performance in independently evolved muscles raises the possibility that the fastest contraction rates measured by Mead et al. represent a maximum speed limit for all vertebrate muscles.

Any technical failure in a racecar most likely will slow it down, while the same failure in a robustly engineered family car may not be so noticeable. Similarly in superfast muscle many cellular and molecular systems need to perform maximally. Therefore by understanding how these extreme muscles work, we also gain a better understanding of how normal muscles contract.
DOI: https://doi.org/10.7554/eLife.29425.002

and mate attraction (*Collins, 2004*). SFMs in the Oyster toadfish swimbladder and rattlesnake tail-shaker both set the fundamental frequency of the produced sound by rhythmically contracting the swimbladder and shaking tail rattles respectively (*Rome et al., 1996*). Other muscles have been suggested to be of the superfast phenotype, such as some extraocular or limb muscles (*Fuxjager et al., 2016*), but non-isometric mechanical tests are lacking to classify them as such. Taken together, SFMs seem a commonly occurring muscle phenotype that interestingly so far has been established only in motor systems involving sound production and control.

Force modulation in vertebrate skeletal muscles is precisely timed via the highly conserved excitation-contraction coupling (ECC) pathway that consists of several sequential steps whose individual kinetics affect the maximally attainable force modulation speed. In brief, firing of a motor neuron first triggers the release of intracellular calcium ions ($Ca^{2+}$) stored in the sarcoplasmic reticulum (SR). Subsequent binding of $Ca^{2+}$ to troponin in sarcomeres triggers the exposure of binding sites for the motor protein myosin along actin filaments, allowing the cyclical binding and unbinding of myosin motor head domains to actin, forming actomyosin crossbridges that generate force. Finally, force decreases when SR $Ca^{2+}$-ATPases (SERCA) pump $Ca^{2+}$ back into the SR, consecutively lowering the cytoplasmic free $Ca^{2+}$ concentration ($[Ca^{2+}]_i$) and returning thin filament inhibition thus preventing further crossbridge formation. Extensive study of SFMs in the oyster toadfish (*Opsanus tao*) swimbladder has demonstrated that no single step in the ECC pathway is rate-limiting, but that multiple hallmark traits have adapted to allow superfast cycling rates. In swimbladder SFMs, a far greater proportion of cellular volume is dedicated to SR compared to locomotory muscles, which increases the number and density of SERCA pumps (*Rome et al., 1999*). Furthermore small, ribbon-like myofibrils (*Appelt et al., 1991*) greatly reduce diffusion distances for $Ca^{2+}$. Together, these adaptations lead to the shortest $[Ca^{2+}]_i$ transient times observed in any muscle (*Rome et al., 1996*). Additionally, the rate of actomyosin crossbridge detachment in SFM of the toadfish swimbladder is extremely fast, which is necessary to ensure a rapid force drop after the return of actin filament inhibition (*Rome et al., 1999*). Whether these hallmark ECC pathway adaptations are shared with avian (songbird syrinx) and mammalian (bat larynx) SFMs are currently unknown.

In vertebrates, kinetic tuning of the actomyosin crossbridge cycle for different energetic and biomechanical requirements is accomplished largely via the differential expression of specific myosin heavy chain (*MYH*) genes each with unique kinetic properties (*Bottinelli et al., 1991*). Because of its thoroughly characterized phylogeny (*Desjardins et al., 2002*) and a demonstrated role in force generation (*Schiaffino and Reggiani, 2011*) the *MYH* gene family is well suited to shed light onto the evolutionary origin and timing of SFM phenotypes. However, *MYH* characterization and expression are currently not known for any SFM. Bat and songbird SFM are excellent models to (i) determine whether SFMs share a common evolutionary origin and to (ii) improve our understanding of muscle function, because sequenced genomes (*Warren et al., 2010*) and identified neural substrates for quantifiable, learned vocalizations exist in some of these taxa (*Fee and Scharff, 2010*; *Brainard and Doupe, 2013*; *Moss and Sinha, 2003*).

Interestingly, functional trade-offs associated with the above ECC adaptations do appear to be common to all SFMs. The fast actomyosin detachment rate of SFM in the toadfish swimbladder, in the absence of a commensurate increase in the rate of crossbridge formation, reduces the proportion of myosin motors actively bound to actin at a given time, and thus force and power production (*Rome et al., 1999*). Furthermore, the increased volumetric allocation for required SR and mitochondria leave less space for contractile machinery, further reducing volume-specific force and power (*Rome and Lindstedt, 1998*). Consequentially, the SFMs in swimbladder and tailshaker trade force for speed, constraining movement to very low masses at low efficiency (*Rome et al., 1999*; *Rome and Lindstedt, 1998*; *Young and Rome, 2001*). SFMs in bat, songbird and toadfish, which produce work up to similar cycling limits (around 200 Hz), develop very similar low force (10–20 kN/m$^2$) (*Rome, 2006*; *Elemans et al., 2004*, *2008*, *2011*; *Rome et al., 1996*). Because of the bias for the SFM phenotype to have evolved in sound production systems we asked whether this apparent functional convergence in SFM force profiles is (1) the result of selective pressures common to motor control of sound production and modulation, or (2) due to constraints inherent to the otherwise conserved architecture of synchronous vertebrate sarcomeric muscle, or (3) both?

Here, we test if SFMs share a common evolutionary origin by characterizing *MYH* gene expression in the SFM of zebra finch (*Taeniopygia guttata*) syrinx and Daubenton's bat (*Myotis daubentonii*) larynx. The employment of evolutionarily and ontologically distinct *MYH* genes suggests separate evolutionary origins of SFM. Furthermore, by quantifying cellular morphometry and [Ca$^{2+}$]$_i$ signal transduction in all known SFM, we show that all SFMs share hallmark adaptations in the ECC pathway and have identical intracellular calcium dynamics at their operating temperatures *in vivo*. Our data suggest that SFMs converge at a maximum speed allowed by fundamental constraints in vertebrate synchronous muscle architecture. This implies that motor control of complex acoustic communication is fundamentally limited by synchronous vertebral muscle architecture.

## Results

### Myosin Heavy Chain gene presence in SFM phenotypes

To test the hypothesis that SFMs in the bat larynx and songbird syrinx share a common evolutionary origin (*Figure 1*), we used sequenced genomes of zebra finch (*Warren et al., 2010*) and little brown bat (*Myotis lucifigus*), an echolocating bat in the same genus as Daubenton's bat, to identify the dominant *MYH* genes expressed in their SFMs. Among known *MYH* genes, orthologs of human *MYH13* are prime candidates for superfast motor performance (*Bloemink et al., 2013*). In humans and other mammals, *MYH13* and five other *MYH* genes (*MYH8*, *MYH4*, *MYH1*, *MYH2*, *MYH3*) are linked head-to-tail in what is known as the fast/developmental cluster (*Desjardins et al., 2002*), which arose from multiple gene duplication events, the first believed to have occurred in an early tetrapod (*Ikeda et al., 2007*). Here, we identified the orthologous fast/developmental *MYH* gene clusters in syntenic regions of zebra finch and little brown bat genomes (*Figure 2a,b*). To place putative superfast bird and bat *MYH* genes in the context of well-characterized vertebrate fast myosin evolution, we additionally used human MYH13 and the slow/cardiac MYH7 motor domain amino acid sequence to identify predicted *MYH* genes by BLASTp in the large flying fox (*Pteropus vampyrus*) (*Kersey et al., 2016*), chicken (*Gallus gallus*) (*International Chicken Genome Sequencing Consortium, 2004*), Burmese python (*Python molurus*) (*Castoe et al., 2013*), clawed frog (*Xenopus tropicalis*) (*Nasipak and Kelley, 2008*), and torafugu puffer fish (*Takifugu rubripes*) (*Aparicio et al.,*

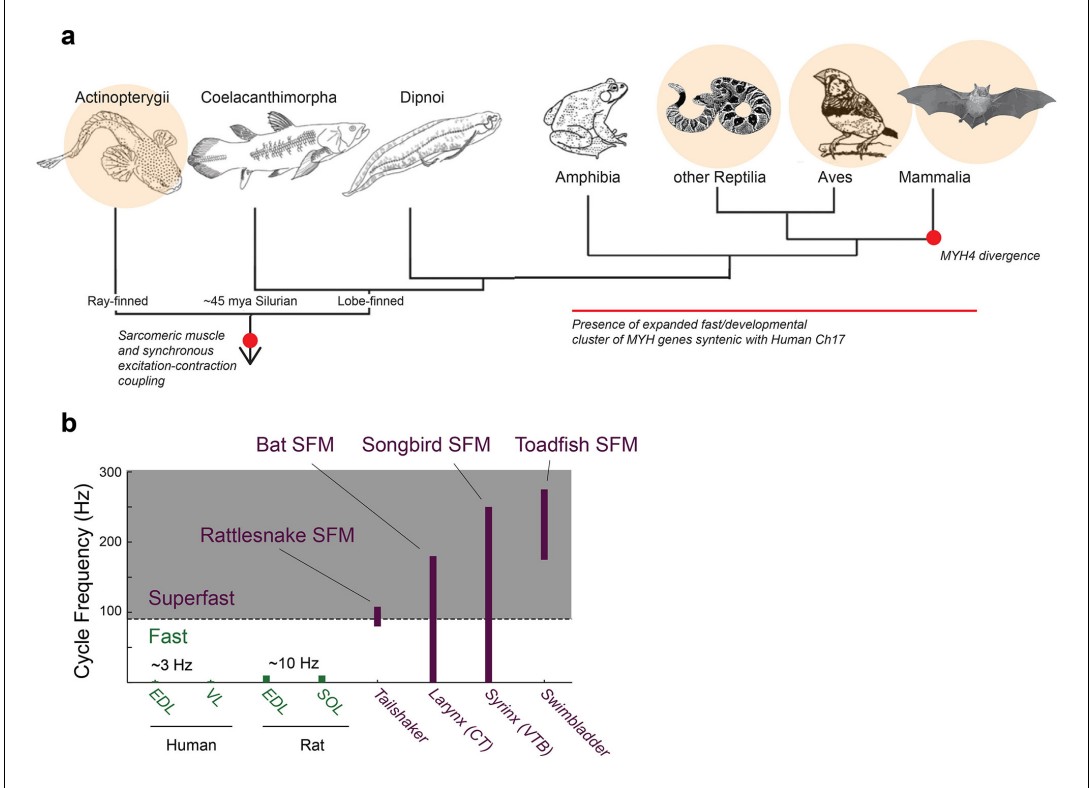

**Figure 1.** Superfast muscles are present in multiple vertebrate lineages. (**a**) Distribution of identified superfast muscles (SFMs, beige disks) in vertebrates relative to key evolutionary events: excitation-contraction coupling, and myosin heavy chain (*MYH*) gene evolution. Modified after (*Bass, 2014*). (**b**) Ranges of *in vivo* cycling frequency. SFM (labeled purple) has been defined as synchronous muscle capable of producing work at cycling frequencies in excess of 90 Hz (grey box) (*Rome, 2006*). Bat, songbird, and toadfish SFM approach or exceed 200 Hz. CT, cricothyroid; VTB, *m. tracheobronchialis ventralis*; EDL, *m. extensor digitorum longus*; SOL, *soleus*; VL, *vastus lateralis*.
DOI: https://doi.org/10.7554/eLife.29425.003

*2002*). Phylogenetic analysis of myosin rod amino acid sequence revealed orthologs of *MYH13* in syntenic genomic regions of all mammalian and avian species included in our analysis corroborating previous studies (*Desjardins et al., 2002*; *Nasipak and Kelley, 2008*). We also found further lineage-specific expansions of fast/developmental clusters (*Figure 2b*), though gene convergence events complicate precise phylogenetic reconstruction of more recently diverging genes (*Desjardins et al., 2002*). Some ray-finned fishes, including torafugu, only have a single *MYH* gene in duplicated genomic regions syntenic to the tetrapod fast/developmental cluster (*Ikeda et al., 2007*), supporting the view that the duplication event at this locus, which gave rise to an ancestral *MYH13* postdated the tetrapod/ray finned fish last common ancestor. Interestingly, laryngeal muscles of the frog, which power vocalizations in the range of 70 Hz, express a *MYH* gene (*MyHC-101d*) that does not appear to have descended from the ancestral fast/developmental locus (*Nasipak and Kelley, 2008*). The python possesses an ortholog of *MYH3* (*LOC103063479*) at the expected 5' end of the cluster, but the presence of a 3' flanking *MYH13* ortholog is unknown due to the presence of a scaffold end.

To test the hypothesis that *MYH13* orthologs are expressed in bat laryngeal and zebra finch syringeal SFMs, we first used a panel of MYH13-specific commercial and custom-made antibodies (See Materials and methods). Surprisingly, we did not find immunoreactivity against MYH13 in bat laryngeal SFM (*Figure 3a*). However, a fast myosin antibody (See Materials and methods), which binds gene products of *MYH 1*, *2*, and *4*, showed immunoreactivity. In agreement with that, sequenced amplicons of qPCR analysis revealed that bat SFM laryngeal muscle is highly enriched for the mammal-lineage-specific *MYH4* transcript, which codes for a locomotory myosin also known as MyHC-2B (*Figure 3b*). In male zebra finch syrinx SFM, multiple attempts at immunostaining were unsuccessful

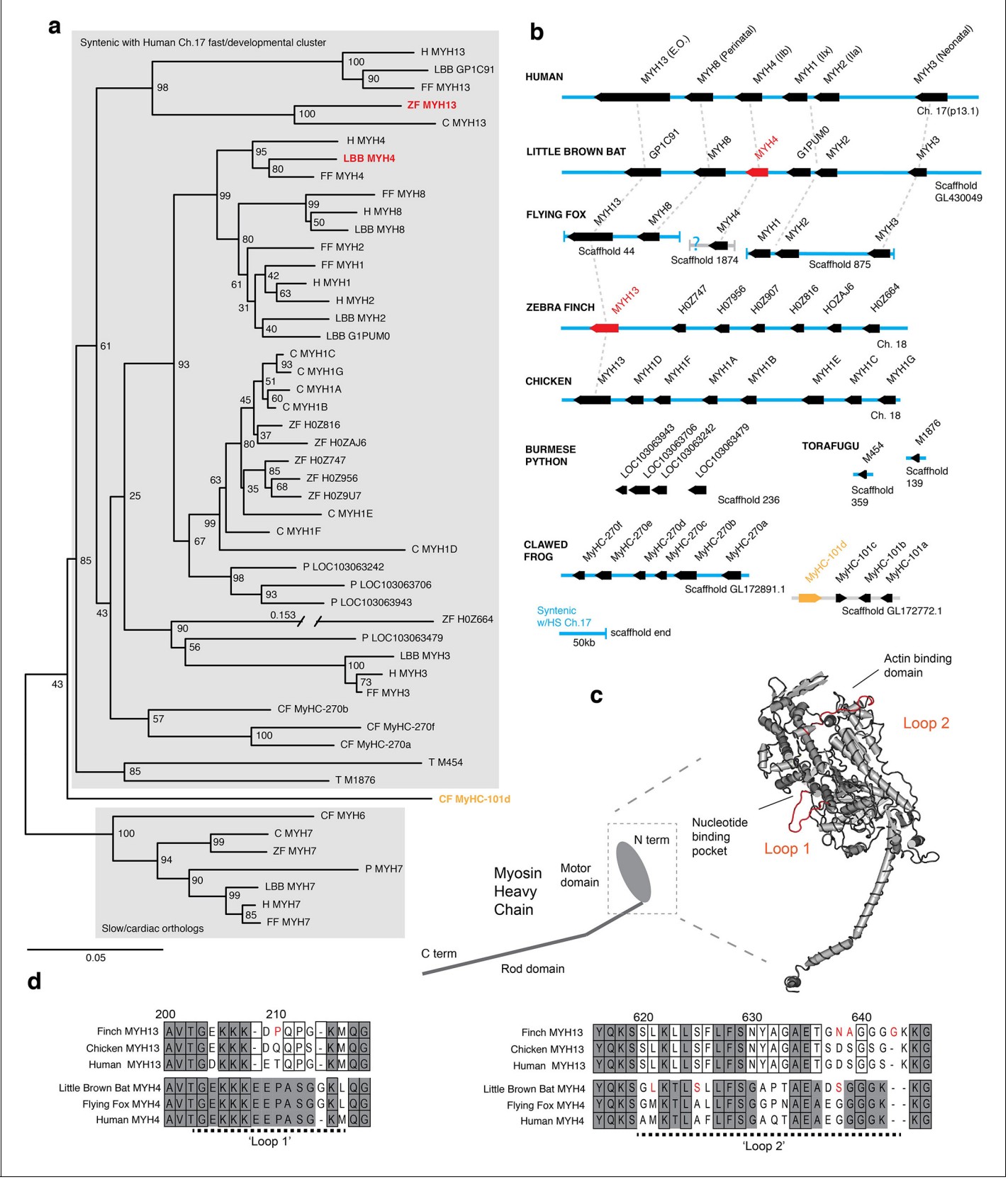

**Figure 2.** Mammals and avians possess orthologous clusters of myosin heavy chain genes. (a) Neighbor-joining tree of chicken (C), clawed frog (CF), flying fox (FF), human (H), little brown bat (LBB), Burmese python (P), torafugu (T), and zebra finch (ZF) predicted myosin heavy chain rod domain amino

*Figure 2 continued on next page*

*Figure 2 continued*

acid sequence from genomic regions syntenic with the human fast/developmental cluster on chromosome 17. Also included are representative cardiac genes, as well as a fast laryngeal myosin gene from the clawed frog (yellow). MYH expressed in bat and finch SFM indicated in red. Human non-muscle *MYH9* was used to root the tree (not shown). Branch lengths shown are derived from a maximum likelihood analysis of aligned amino acid sequence. Bootstrap values from a 1000-replicate analysis are given at nodes in percentages. (b) Relative local positions of predicted *MYH* genes in genomes of the above species. Synteny with human Ch17 is shown in blue. Red and yellow color-coding as in (a). Orthologs indicated by vertical dotted lines. (c) Homology model of human *MYH3* myosin motor domain indicating position of loop subdomains and the nucleotide binding pocket. (d) Hypervariable surface loops of *MYH13* and *MYH4* motor-domains that likely influence actomyosin crossbridge kinetics. Grey shading indicates conservation with/ among *MYH4* orthologs. Outlines indicate conservation with/among *MYH13* orthologs. Substitutions and insertions unique to SFM species in red. Horizontal black dashed lines indicate 'Loop 1' and 'Loop 2' subdomains.

DOI: https://doi.org/10.7554/eLife.29425.004

(see Materials and methods), but sequenced amplicons of qPCR analysis unambiguously showed that it was highly enriched for the *MYH13* ortholog transcript (*Figure 4a*). Only one other myosin gene (*H0Z747*) was expressed and we could not distinguish whether this myosin was expressed in different cells, as suggested by observation of fast myosin antibody staining in starlings (*Uchida et al., 2010*) and zebra finches (*Figure 4—figure supplement 1*), or co-expressed in the same cells. Extraocular muscle expressed *MYH13* at the same level as syringeal SFM (p=0.06, Kruskal Wallis), and also four additional *MYH* genes from the fast/developmental cluster (*Figure 4a*).

To identify possible species-specific adaptations to *MYH4* in the echolocating bat, and to *MYH13* in the zebra finch that could affect crossbridge kinetics, we aligned predicted motor domain amino acid sequence from these genes to their orthologs in the genomes of human, large flying fox and chicken, respectively, focusing on two hypervariable regions identified as potential modifiers of crossbridge kinetics (*Figure 2c,d*). Loop one is situated near the nucleotide binding pocket, and its flexibility is thought to influence the rate of ADP dissociation, and thus is a potentially critical region

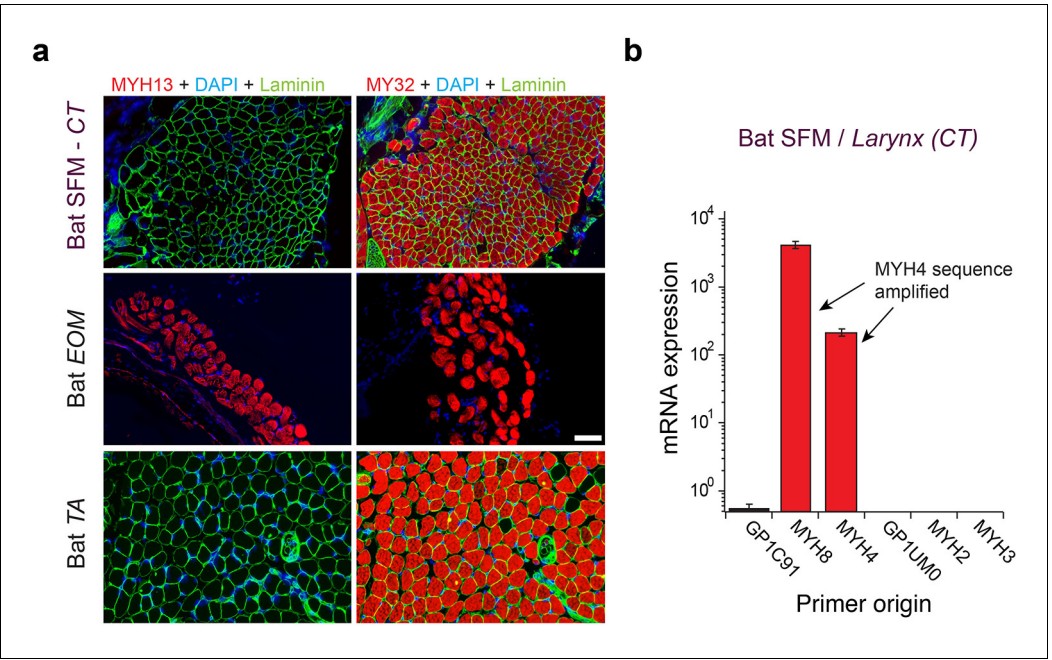

**Figure 3.** Mammalian superfast muscles are enriched with *MYH4* ortholog. (a) Immunohistochemistry labeling shows absence of MYH13 (left column) in Daubenton's bat larynx SFM (top) with positive control in extraocular muscle (EOM) (middle) and negative control in body skeletal muscle (*m. tibialis anterior*; TA) (bottom). Fast twitch MYHs (right column) are present in SFM, EOM and TA muscles. (b) qPCR analysis identifies *MYH4* (MyHC-2b) to be dominating in bat SFM. Both *MYH4* and *MYH8* primer pairs amplified *MYH4* mRNA as confirmed by cDNA sequencing.

DOI: https://doi.org/10.7554/eLife.29425.005

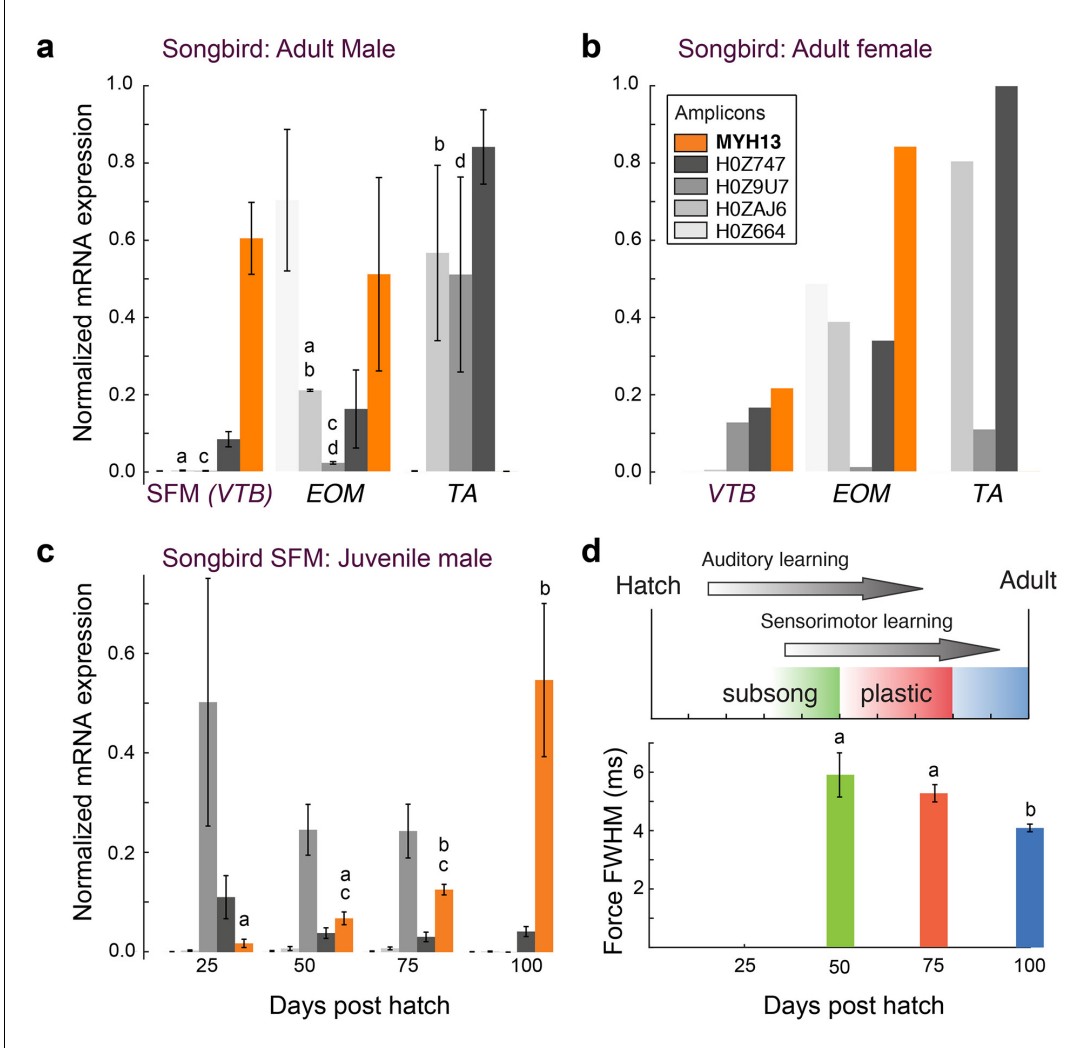

**Figure 4.** Avian superfast muscles are enriched for the myosin *MYH13* ortholog. (**a**) qPCR analysis of syringeal SFM of adult male (N = 3) and (**b**) female (N = 3, pooled) zebra finch show high enrichment for the *MYH13* ortholog in singing males. Legend for color code of bars in b. Unshared lowercase letters a-d indicate post-hoc differences with significance level p<0.05. Fast myosin (MY-32) immunohistochemistry results are shown in *Figure 4—figure supplement 1*. (**c**) During the sensorimotor phase of vocal imitation learning in male zebra finches *MYH13* is significantly upregulated (N = 3 for each timepoint). Legend for color code of bars in b. Unshared lowercase letters a-c indicate differences with significance level p<0.05. (**d**) Syrinx SFM significantly increases their speed during song development (N = 4 for each timepoint). FWHM; full width at half maximum.
DOI: https://doi.org/10.7554/eLife.29425.006

The following figure supplement is available for figure 4:

**Figure supplement 1.** Zebra finch SFM contains two fiber type populations.
DOI: https://doi.org/10.7554/eLife.29425.007

for the rapid crossbridge detachment seen in SFM (*Ikeda et al., 2007*). The only non-synonymous sequence difference between little brown bat and human MYH4 in loop 1, an insertion of a glycine at position 214, is also shared with the large flying fox. Although human MYH13 is known to possess rapid ADP dissociation (*Bloemink et al., 2013*), the zebra finch ortholog differs from those of both human and chicken by the substitution of a proline at position 210. Loop two potentially affects maximum shortening velocity via influence on actin-myosin binding (*Lorenz and Holmes, 2010*). Both zebra finch *MYH13* and bat *MYH4* possess amino acid substitutions in loop two and in the case of the finch the insertion of a glycine at position 643 not shared by chicken, flying fox, or human orthologs.

## Associations of MYH expression and motor performance with singing behavior

To further explore the relationship between *MYH* gene expression and muscle speed, we next investigated properties of songbird SFM in three groups of zebra finches that differ with respect to song production: juvenile males during the song learning phase singing juvenile, variable song, adult males singing adult, invariable song, and females that do not sing. All three groups produce different types of mostly unlearned calls. Female syringeal SFMs have two-fold slower twitch kinetics than male syringeal SFMs (*Elemans et al., 2008*), and expressed more than five times less *MYH13* compared to male syringeal SFMs (*Figure 4b*). Juvenile male songbirds develop their vocal behavior by sensory-guided motor practice in a process bearing many parallels to human speech acquisition (*Fee and Scharff, 2010*; *Brainard and Doupe, 2013*). Over the 75-days long sensorimotor phase young zebra finches produce vocalizations that slowly change in acoustical parameters (*Tchernichovski et al., 2001*). During this period, the variability of acoustic output decreases and accuracy increases, which is attributed to increasingly precise timing of vocal motor pathway activity (*Ölveczky et al., 2011*). To see whether these vocal changes are associated with changes in SFM performance, we first quantified *MYH* expression in syringeal SFM over song ontogeny. Of all *MYH* genes, only *MYH13* changed significantly with age (p=0.02, KW-test) and was significantly upregulated progressively (p<0.05, Tukey-Kramer post hoc) in juvenile male syringeal SFM (*Figure 4c*). Second, we found that muscle twitch duration significantly decreased progressively (p<0.05, Wilcoxon-signed rank tests) during the sensorimotor period of vocal learning (*Figure 4d*). Taken together, these results suggest that in zebra finch syrinx SFM levels of *MYH13* expression are negatively associated with twitch duration and thus positively with muscle speed.

## Morphometric cellular adaptations converge in SFM

Because the above results established that SFMs do not share a common evolutionary origin, we next investigated whether SFM share adaptive mechanisms to ECC pathway traits. We first examined whether cellular adaptations for rapid calcium transients are present across all known SFMs. Representative transmission electron-microscopy cross-sections of SFMs in zebra finch syrinx, bat larynx, toadfish swimbladder and rattlesnake tailshaker showed strikingly more SR compared to intraspecific skeletal muscles (*Figure 5a*). Based on transmission electron-microscopy images, we quantified volumetric percentages of sarcoplasmatic reticulum, mitochondria and myofibrils in all four SFM and intraspecific skeletal muscles (*Figure 5b*). Zebra finch syrinx SFMs contained 15 ± 1, 22 ± 2 and 50 ± 2% of SR, mitochondria and myofibrils, respectively (N = 3) and transmission electron-microscopy images resembled those of dove (*Elemans et al., 2006*) and oilbird (*Suthers and Hector, 1985*) syrinx muscles. Bat larynx SFMs contained 24 ± 2, 31 ± 4 and 34 ± 2% of SR, mitochondria and myofibrils, respectively (N = 2) (*Figure 5b*). SR volume was ≥15% and SR per myofibril ratio was ≥30% for all SFMs and significantly higher (p<0.05, KW test) compared to intraspecific skeletal muscles in zebra finch, bat and rattlesnake. Thus the morphometric cellular adaptation of larger SR volume to increase signal transduction (*Rome, 2006*; *Rome and Lindstedt, 1998*) is a hallmark adaptation present in all SFM.

## In vivo $[Ca^{2+}]_i$ Signal Transduction Dynamics Converge in SFM

We investigated whether $[Ca^{2+}]_i$ transients in songbird syrinx and bat larynx SFM attain similar extreme kinetics as toadfish swimbladder and rattlesnake tailshaker SFM (*Rome et al., 1996*). We measured force and real-time $[Ca^{2+}]_i$ dynamics in zebra finch syrinx and toadfish swimbladder SFM as a function of temperature (*Figure 6*). At their respective operating temperatures of 39°C and 25°C, the full width at half maximum (FWHM) values of calcium transients for zebra finch syrinx and toadfish swimbladder SFMs did not differ significantly (p=0.54, one-tailed paired t-test) and measured 1.98 ± 0.65 ms (N = 4) and 1.91 ± 0.69 ms (N = 2), respectively. The $FWHM_{[Ca2+]i}$ in rattle snake tailshaker SFM was 1.5 ms at 35°C (*Rome et al., 1996*). Taken together, the $FWHM_{[Ca2+]i}$ values for SFMs in zebra finch, toadfish, and rattlesnake converged to 1.5–2.0 ms at their operating temperatures (*Figure 6b*). In toadfish SFMs, the magnitude of $[Ca^{2+}]_i$ transients and ATP-usage reduces from the first to consecutive twitches that make up the vast majority of the boatwhistle call (*Harwood et al., 2011*). If the FWHM of $[Ca^{2+}]_i$ transients also decreases from first to consecutive twitches, our above comparison would underestimate the speed of the toadfish SFM transient.

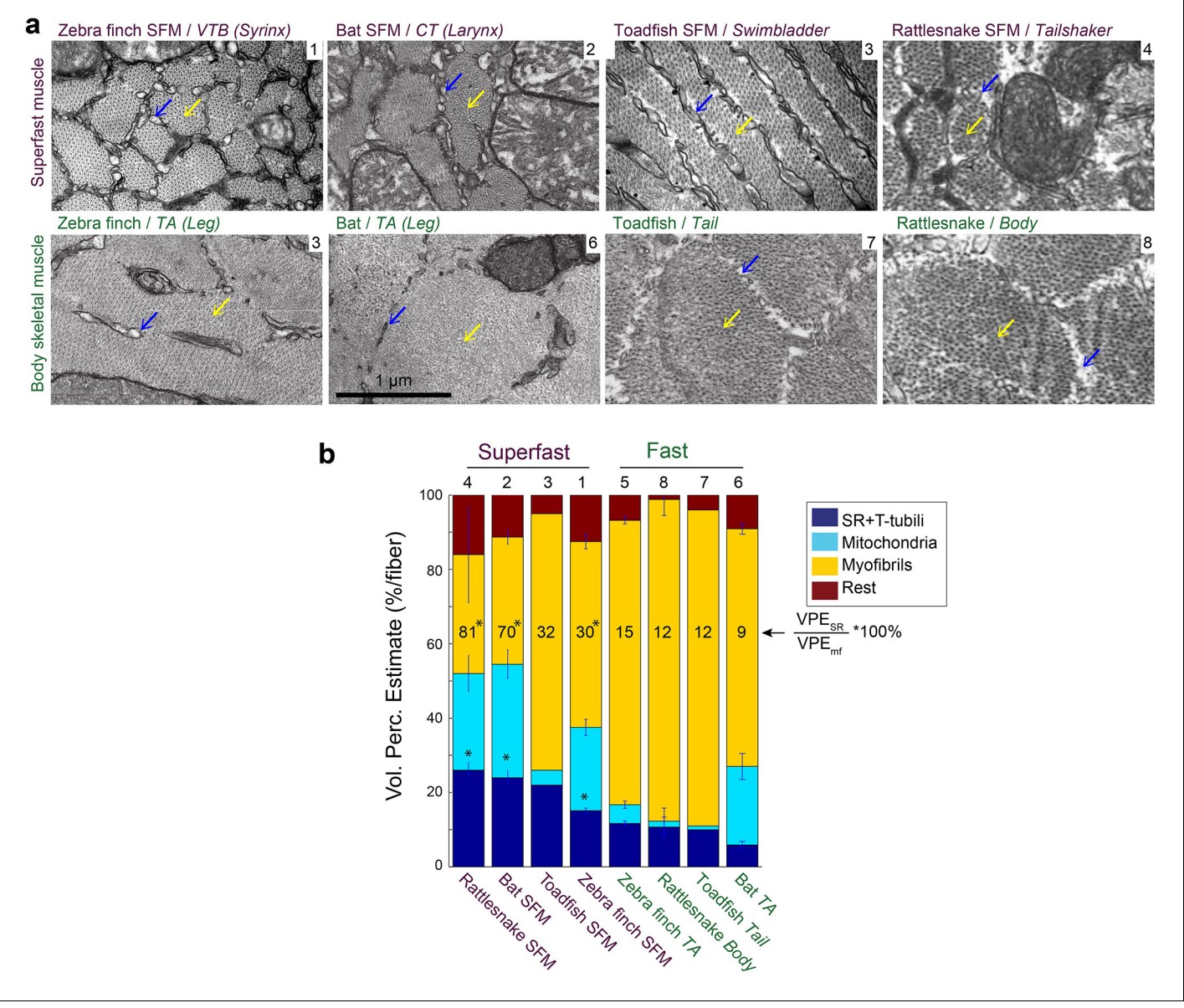

**Figure 5.** Vertebrate superfast muscles share morphometric cellular adaptations. (a) Representative transmission electron microscopy images of zebra finch, Daubenton's bat, Oyster toadfish and Diamondback rattlesnake SFMs (upper row) and intraspecific skeletal muscles (lower row). Blue arrows indicate SR, yellow arrows myofibrils. Rattlesnake muscle images adapted from *Schaeffer et al., 1996*. (b) Volumetric percentage estimates (VPE) of myofiber composition (see Materials and methods) show that all SFM have $VPE_{SR} \geq 15\%$, and SR per myofibril ratio $\geq 30\%$. Both $VPE_{SR}$ and SR per myofibril ratio are significantly higher compared to fast intraspecific skeletal fibers for zebra finch, bat and rattlesnake (*, $p < 0.05$). Data presented in descending order of SR per myofibril ratio. The numbers 1–8 above the bars refer to images in panel (a).
DOI: https://doi.org/10.7554/eLife.29425.008

Therefore, we quantified $[Ca^{2+}]_i$ kinematics as a function of twitch number in toadfish and zebra finch SFM at operational temperatures. While the magnitude of the $[Ca^{2+}]_i$ transient decreased from first to consecutive twitches confirming earlier work on toadfish SFM (*Harwood et al., 2011*), the FWHM of the $[Ca^{2+}]_i$ transients did not change as a function of twitch number (*Figure 6—figure supplement 1*). In zebra finch SFMs, there was no significant magnitude (KW, $p > 0.05$) or FWHM decrease (KW, $p > 0.05$) of $[Ca^{2+}]_i$ transients as a function of twitch number (*Figure 6—figure supplement 1*). Because of limited tissue availability of bat laryngeal SFMs, we used an *in vitro* assay to quantify calcium uptake and release rates of homogenized SR vesicles as a proxy for whole cell behavior (see

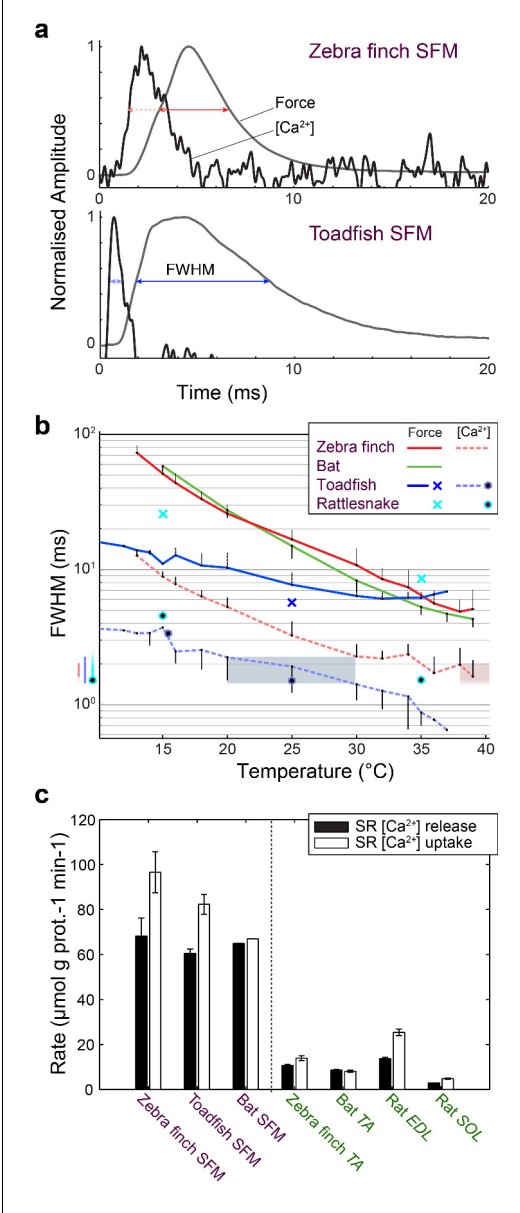

**Figure 6.** Superfast muscles have convergent elevated myoplasmic calcium transient dynamics. (a) Representative force development and myoplasmic calcium concentration ([$Ca^{2+}$]$_i$) transients in songbird and toadfish SFMs. (b) Force and [$Ca^{2+}$]$_i$ transient full-width-at-half-maximum (FWHM) values as a function of temperature. [$Ca^{2+}$]$_i$ FWHM values converge to 1.5–2.0 ms for all SFMs (color left of ordinate) at their operating temperatures (shaded areas). Filled circle and crosses refer to toadfish and rattlesnake data from ref 6. (c) Calcium loading and unloading kinetics in isolated SR vesicles (see Materials and methods) yield similar values for bat SFMs compared to songbird and toadfish SFMs, and were significantly elevated (p<0.01, ANOVA) 5–10 times compared to intraspecific skeletal muscle in zebra finch and bat.

DOI: https://doi.org/10.7554/eLife.29425.009

*Figure 6 continued on next page*

Materials and methods). This technique yields comparable values for SFMs in zebra finch, toadfish and bat, which were significantly elevated (p<0.01, ANOVA) 5–10 times compared to intraspecific skeletal muscle and also compared to the comparable rat fast twitch *m. extensor digitorum longus* (*Figure 6c*, *Supplementary file 1A*). These data thus suggest that bat larynx SFMs have equally fast calcium dynamics as zebra finch syrinx and toadfish swimbladder SFMs. This in turn implies that all established SFMs have similar speeds of calcium dynamics at their operating temperatures *in vivo*. In conclusion, all SFM phenotypes demonstrate hallmark cellular adaptations and functionally convergent calcium transduction dynamics consistent with superfast cycling rates.

## Discussion

We identified the *MYH* genes that power superfast motor performance in songbird syrinx and bat larynx muscle: zebra finch syrinx SFMs predominantly express the avian ortholog of *MYH13*, which encodes superfast MyHC-sf (aka extraocular MyHC-eo), while Daubenton's bat larynx SFMs express an ortholog of the mammalian locomotory *MYH4* (MyHC-2b). Our analysis is consistent with earlier findings (*Desjardins et al., 2002*; *Ikeda et al., 2007*) that both *MYH13* and *MYH4* were the result of gene duplication events that occurred in tetrapods, and would, therefore, have no ortholog in the ray-finned fish, the taxon to which the toadfish belongs. Which *MYH* gene is expressed in the rattlesnake tailshaker SFM remains unknown, but no *MYH13* ortholog was found in the Burmese python, which could mean that the gene was lost, or the genome is incomplete and the fast/developmental cluster was truncated by the end of a scaffold. Since extremely fast crossbridge kinetics are necessary, though not sufficient, for superfast performance (*Rome, 2006*), and these are set largely by *MYH* gene expression in vertebrates (*Bottinelli et al., 1991*), the parsimonious conclusion is that the SFM phenotype found in the sound-producing organs of bats, songbirds, and toadfish descends from different evolutionary events.

We further show that all SFMs share hallmark ECC pathway adaptations to speed up signal transduction. First, we establish that all SFMs converge upon identical speed of *in vivo* calcium dynamics, which can be explained by increased SR per myofibril volume ratio. Additional mechanisms to increase the speed of [$Ca^{2+}$]$_i$ transients are also possible and include (1) adaptations that

*Figure 6 continued*

The following figure supplement is available for figure 6:

**Figure supplement 1.** Calcium transient speed is stable over multiple stimulations.

DOI: https://doi.org/10.7554/eLife.29425.010

affect the function of ryanodine receptors or the SERCA pumps (*Nelson et al., 2016*), (2) faster $Ca^{2+}$ detachment from troponin due to differential expression of Troponin C genes, or (3) the employment of calcium-sequestering proteins such as parvalbumin (*Rome, 2006*) that have already been shown to play an important role in toadfish (*Harwood et al., 2011*; *Tikunov and Rome, 2009*). The potential contribution of these mechanisms needs further investigation across all SFMs. Second, actomyosin crossbridge cycling with a higher than normal detachment rate also appears to be a hallmark of SFM, as evidenced by consistently low developed isometric tensions in all cases (*Rome, 2006*; *Elemans et al., 2004*, *2008*, *2011*). The expression of a *MYH13* ortholog in zebra finch syrinx SFM is consistent with the human protein's superfast kinetics (*Bloemink et al., 2013*), but if and how the observed predicted protein sequence differences between zebra finch and human *MYH13* orthologs affect crossbridge kinetics remains unknown. Similarly, further investigation is required to understand how the normally much slower and higher force locomotory MYH4 realizes superfast crossbridge kinetics in bats. The insertion of a second glycine in loop one may be a potential mechanism to increase crossbridge cycling by altering mechanical flexibility of the loop. However, this insertion is shared by the large flying fox that does not produce calls at high repetition rates, which casts doubt on the insertion being a SFM phenotype specific adaptation. Importantly, superfast crossbridge kinetics may also be the result of adaptations independent of *MYH* genes, so that potential modifiers of crossbridge kinetics, such as myosin light chain genes and phosphorylation state (*Sweeney and Stull, 1990*), also warrant further study.

Both mechanistic as well as functional characteristics are convergent in SFMs. Previous observations show that SFMs from diverse taxa have remarkably similar maximum performance characteristics (force and cycling frequency) (*Rome, 2006*; *Elemans et al., 2004*, *2008*, *2011*; *Rome et al., 1996*; *Moon et al., 2002*). Here, we demonstrate that the similarity extends to the speed of signal transduction itself within the ECC pathway (calcium release and reuptake), and to mechanisms of attaining those speeds (volume and distribution of SR). Because we establish SFM's divergent origin, we propose that these similarities must be attributed to convergent evolution. There is good evidence for selective pressures on high cycling speeds in the motor systems in which SFMs have evolved: female canaries prefer faster trill rates (*Drăgănoiu et al., 2002*), echolocating bats could produce much higher call rates before introducing call-echo ambiguity to their sensory system (*Elemans et al., 2011*) and in toadfish call fundamental frequency, which is set 1:1 by muscle speed (*Elemans et al., 2014*), positively correlates with male fitness (*Amorim et al., 2010*). Thus faster muscles would be expected to evolve.

What mechanisms set the maximum speed to power motion in vertebrates? In toadfish swimbladder SFMs, a trade-off between force-generating capacity and speed is well-established and results from two constraints related to (i) crossbridge dynamics and (ii) increased ECC pathway requirements (*Rome, 2006*). The consequence of fast relaxation rate is that a lower proportion of myosin motors is bound to actin at a given time reducing the developed force and, by extension, the specific force of the myofibril as a whole (*Rome et al., 1999*). While detachment rates of intact toadfish SFM fibers (*Rome et al., 1999*) and isolated MYH13 (*Bloemink et al., 2013*) are very high, attachment rates approximate that of locomotory muscle, and are likely limited by diffusion of the unbound myosin head (*Rome et al., 1999*). Furthermore, the necessity for very rapid ECC signal transduction requires more cell volume dedicated to increased demands on calcium dynamics and ATP production, limiting space for myofibrils, further diminishing force generation per unit volume of muscle (*Rome and Lindstedt, 1998*). Importantly, additional SR and mitochondria are likely to affect the muscle cell's viscoelastic properties. The ultimate limit to how much force can be traded for speed, while the muscle still maintains the ability to do external work, is dictated by the minimum force required to overcome unavoidable viscous losses associated with shortening. Taken together, these observations support the view that SFMs converge at a maximum speed allowed by fundamental constraints in vertebrate synchronous muscle architecture.

Interestingly, only arthropods have pushed the envelope of force/speed tradeoffs further than vertebrate SFMs. In insect asynchronous flight muscle force cycling is uncoupled from calcium

cycling, minimizing force loss and allowing for high power production at similar cycling speeds to SFM (*Syme and Josephson, 2002*). However, this increase in power at high speeds is at the great cost of sacrificing precise temporal control of contraction (*Syme and Josephson, 2002*). The cicada's synchronous tymbal muscles cycle at over 500 Hz (*Josephson and Young, 1985*), but it is unknown if mass reduction or adaptations to the ECC account for this extreme performance. In the much larger vertebrates, the low forces developed by SFMs restrict them to low-mass systems as found in sound production and modulation where precise control is essential.

Myoblast lineage and postnatal motor activation patterns and exercise can play crucial roles in the development of muscle groups or allotypes (*Spencer and Porter, 2006*) and can alter myosin expression in craniofacial (*Rhee et al., 2004*; *Brueckner et al., 1999*) and body (axial) muscle (*Pette and Vrbová, 1985*). Our discovery of *MYH13* ortholog expression in avian syrinx SFM corroborates earlier developmental studies (*Noden et al., 1999*; *Noden and Francis-West, 2006*) and places syringeal myogenic precursor cells in the craniofacial muscle group, because this muscle group can uniquely express *MYH* genes, such as *MYH13*, *15* and *16*, that have never have been found in axial somatic muscle cells. In juvenile songbirds, *MYH13* expression was progressively upregulated during the sensorimotor period of vocal learning and positively associated with muscle speed (*Figure 4c,d*). This observation raises the interesting questions whether *MYH13* upregulation and increase of muscle speed in songbird syrinx SFMs is causally related and due to a set developmental program, hormonal influences, muscle exercise in the form of increased neural stimulation, or some combination of these factors. Interestingly, the SFM in the larynx of juvenile bats may also increase speed during the first 30 days after birth as suggested by an increase in frequency modulation speed of echolocation calls (*Moss et al., 1997*). In other motor systems, neural stimulation can drive *MYH* expression patterns: neural activity associated with optokinetic and vestibulo-ocular reflexes stimulates particularly *MYH13* expression in rat extraocular muscle (*Brueckner et al., 1999*; *Moncman et al., 2011*), and transnervation from cranial nerve X with XII increases speed of laryngeal muscles in dogs (*Paniello et al., 2001*). Because in extraocular muscle MYH13 is found close to the neuromuscular junction (*Briggs and Schachat, 2002*), a possible mechanism could be that both electrical and chemical activation of the motor neuron directly stimulate MYH13 production linked to actetylcholine receptors (*Rubinstein et al., 2004*; *Sanes and Lichtman, 2001*). We speculate that vocal muscle training can be associated with optimizing SFM function. Reversely, our data suggest that, next to neural constraints, SFMs can peripherally constrain the precision to execute skilled motor sequences during birdsong and echolocation call ontogeny.

To conclude, the non-orthology of the dominant myosin heavy chains expressed in SFMs of birds, mammals, and fish indicates that SFMs likely evolved independently in each lineage. However, these independent SFM each employs similar qualitative and quantitative adaptations to ECC, and arrive at near identical maximum performance. Taken together, these results suggest that SFMs operate at the maximum speed allowed by vertebrate synchronous muscle architecture. In each case, SFM evolution involved the same specific set of constraints (high cycling speeds and synchronous control) and allowances (minimal force and power), particular to motor systems associated with sound production. Motor-driven acoustic modulation rates in complex communication are thus fundamentally limited by muscle architecture.

## Materials and methods

### Subjects and tissue collection

All procedures were carried out in accordance with the Danish Animal Experiments Inspectorate (Copenhagen, Denmark).

*Songbird:* All bird data presented were collected in zebra finches (*Taeniopygia guttata* (Vieillot, 1817). All adult birds (older than 100 days) were kept in group aviaries at the University of Southern Denmark, Odense, Denmark, and juvenile zebra finches used for muscle kinetics measurements were raised in separate cages together with both parents and siblings at 13 hr light:11 hr dark photoperiod and given water and food *ad libitum*. The juvenile male zebra finches used for *MYH* mRNA quantification came from the long-term breeding colony at the Freie Universität Berlin, Germany. These animals were kept on a 14 hr light:10 hr dark photoperiod and given water and food *ad libitum*. We dissected the syringeal muscle *tracheobronchialis ventralis* (VTB) on ice immediately after

isoflurane euthanasia (*Elemans et al., 2008*). Extraocular (*m. rectus* and *m. oblique*; EOM), flight (*m. pectoralis*; PEC) and leg (*m. tibialis anterior*; TA) muscles were collected as reference tissues.

*Toadfish:* Adult male oyster toadfish (*Opsanus tau*, Linnaeus 1766) were obtained from the Marine Biological Laboratories (MBL, Woods Hole, MA, USA) and housed individually in 80 × 40×40 cm seawater tanks on constant flow-through of fresh seawater at the Fjord and Bælt field station, Kerteminde, Denmark. Toadfish were euthanized by a blow on the head and double pithing. Superfast swimbladder muscle was isolated bilaterally from the midsection of the swimbladder.

*Bats*: Efforts were focused on Daubenton's bat (*Myotis daubentoni*) where superfast behavior of the anterior portion of the laryngeal muscle *m. cricothyroideus anterior* (ACTM) was previously established (*Elemans et al., 2011*). We obtained permission (Licenses SNS-3446–00001 and NST-3446–00001) to capture four individuals of *Myotis daubentoni* from the Skov- og Naturstyrelsen Inspectorate (Denmark). After isoflurane euthanasia, the ACTM muscle was isolated and extraocular and leg (*m. tibialis anterior*, TA) muscles were collected as reference tissues.

*Rats:* Adult male Sprague Dawley rats (*Rattus norvegicus*) were purchased from the Institute of Biomedicine, Odense University Hospital. The rats were housed in cages with a 12 hr:12 hr light:dark cycle and provided unrestricted access to water and food. The rats were killed by a blow on the head followed by cervical dislocation. *M. soleus* (SOL) and *m. extensor digitorum longus* (EDL) were collected.

## Myosin heavy chain phylogeny

Fast/developmental myosin heavy chain gene clusters were identified in the genomes of *Myotis lucifigus* and *Taeniopygia guttata* by BLASTP using ENSEMBL genome resources (http://useast. ensembl.org/Myotis_lucifugus/Info/Index and http://useast.ensembl.org/Taeniopygia_guttata/Info/Index, RRID:SCR_006773) with the rod domain of human *MYH3* (conserved motor domain/rod junction proline 839 to C terminus) as query sequence. Seven Ensembl genebuild annotated predicted proteins were clustered on forward strand of the zebra finch chromosome 18. Six predicted proteins in bat were similarly clustered on the reverse strand of scaffold GL430049 (*Supplementary file 1B*). The Ensemble annotation for little brown bat *MYH13* was missing exons 1 through 12. Fgenesh gene-finder (hidden markov model based gene structure prediction - http://linux1.softberry.com/berry.phtml?topic=fgenesh&group=programs&subgroup=gfind, RRID:SCR_011928) was used to partially reconstruct the full-length gene, including loop 1 and loop two subdomains, from flanking genomic sequence (*Figure 2*). TBLASTN using the same query sequence returned no additional *MYH* domains in these genomic regions.

Fast/developmental cluster and cardiac orthologs from flying fox, chicken, and clawed frog genomes were identified by BLAST using ensemble genome resources, and NCBI genome, with human *MYH3* and *MYH7* (conserved motor domain/rod junction proline 839 to C terminus) as query sequence. Torafugu sequences were obtained directly from (*Ikeda et al., 2007*). 2D dot-plot analysis of genomic DNA vs. human MYH3 protein sequence was used to identify clawed frog genes described in (*Nasipak and Kelley, 2008*), which had been named using scaffold names from an earlier assembly (*Supplementary file 1B*). Fgenesh gene finder was used to reconstruct MyHC-101d sequence from genomic sequence on scaffold 172772.1.

Predicted protein sequences were downloaded and analyzed using Macvector 14.5 software (Macvector Inc. Apex, NC). Human MYH sequences were obtained from the NCBI protein database (*Supplementary file 1B*). To remove the influence of insertions/deletions, multiple protein sequence alignments of human sarcomeric *MYH*s, along with non-muscle *MYH9,* were performed using Clustal W (*Thompson et al., 1994*) in Macvector on rod domains from the conserved proline at the motor domain/rod junction (839 in MYH3) to the C terminus.

To reconstruct the phylogeny of fast/developmental cluster *MYH* genes from zebra finch, little brown bat, chicken, large flying fox, Burmese python, clawed frog, torafugu and human genomes (*Warren et al., 2010*; *Kersey et al., 2016*; *International Chicken Genome Sequencing Consortium, 2004*; *Castoe et al., 2013*; *Nasipak and Kelley, 2008*; *Aparicio et al., 2002*), alignments were performed on 556-residue regions of the rod domain starting with the conserved alanine (1331 in human *MYH3*). Here full-length rods were not aligned due to a region of unknown sequence in bat *MYH13*. Maximum likelihood trees were generated and branch lengths calculated in Macvector using the Neigbor Joining method (*Saitou and Nei, 1987*) rooted by the non-muscle *MYH9*

(*Figure 2a*, not shown). Bootstrap analysis from 1000 replicates was used to evaluate internal branches.

For the purposes of comparing relevant hypervariable regions within the motor domain, Clustal W alignments were performed on N-terminal motor domain predicted protein sequence, terminating at the conserved most-c-terminal proline (839 in human *MYH3*) from human *MYH13*, little brown bat and flying fox *MYH4*; chicken and zebra finch *MYH13*. The three-dimensional homology model in *Figure 2c* was generated for human MYH3 protein, with scallop myosin head structures (ProteinData Bank code 1kk8) as a template, using the SWISS-MODEL (RRID:SCR_013032) server (*Arnold et al., 2006*) as in (*Bloemink et al., 2013*).

## Immunohistochemistry

Muscle tissue was rapidly dissected and frozen in liquid nitrogen after sacrifice. Immunofluorescence staining was carried out on 5 µm thick frozen sections. After initial washing with PBS three times for 5 min, sections were incubated for 20 min in a 1% solution of Triton X-100 (Mannheim, Germany) in 0.01 M PBS (Roche, Mannheim, Germany) and rinsed in PBS three times. The sections were then incubated in 10% normal goat serum (Life technologies, MD, USA) for 15 min, and were thereafter incubated with a mouse monoclonal anti-myosin (Skeletal, fast) antibody (clone MY-32 (RRID:AB_784724), M4276 (RRID:AB_477190), Sigma-Aldrich, MO, USA), diluted 1:100; a rabbit polyclonal myosin 13/Extraocular myosin (Hinge region) (MP4571 (RRID:AB_2713977), ECM Biosciences, KY, USA), diluted 1:50; and a rat monoclonal anti-Laminin 3 alpha antibody (4H8-2, ab11576 (RRID:AB_298180), Abcam, MA, USA), diluted 1:500 in PBS with BSA for 60 min at 37°C. The skeletal fast antibody MY-32 binds to MYH1, 2 and 4. To our knowledge no existing antibody can reliably distinguish between these myosins. After incubation with antiserum and PBS wash, a new incubation in 10% normal goat serum followed, after which the sections were incubated in goat anti-mouse IgG1, Human ads-TRITC (1070–03, SouthernBiotech, AL, USA), donkey anti-rabbit IgG-TRITC (711-025-152, Jackson ImmunoResearch, PA, USA) and goat anti-Rat IgG Alexa 488 (ab150157, Abcam, MA, USA) diluted 1:300 for 30 min at 37°C. The sections were thereafter washed in PBS and then mounted in Vectashield Mounting Medium (H-1500) with DAPI (Vector Laboratories, CA, USA). Images were taken using Leica DM6000 at University of Pennsylvania Microscopy Core Facility.

We explored six antibodies to identify MYH13 in zebra finch, but all attempts were unsuccessful. Summarizing, we tried to raise one poly- and one monoclonal antibody against a peptide derived from ortholog-specific loop 2 sequence, tested three commercially available antibodies (of which one was used successfully in bats above), and tested one antibody used successfully in rabbits (*Rhee and Hoh, 2008*). All these antibodies were tested by experienced workers and were either unsuccessful or lacked positive identification (i.e. absence of evidence).

## RNA preparation and qRT-PCR analysis

Muscle tissues were rapidly dissected and frozen in liquid nitrogen after sacrifice. Tissues were homogenized in Trizol using an Ultra-turrax homogenizer and total RNA was extracted according to manufacturer´s instructions (Invitrogen). Reverse transcription was performed by incubating 1 µg of total RNA with 0.25 µg of random hexamers (Amersham Pharmacia Biotech) and 0.9 mM dNTPs at 65°C for 5 min, followed by incubation with 1x First Strand Buffer (Invitrogen), 10 mM DTT and 200 units of Moloney murine leukemia virus reverse transcriptase (Life Technologies) at 37°C for 1 hr. For more details see (*Osinalde et al., 2016*). Quantitative PCR was performed using 20 µl reactions containing SYBR Green JumpStart Taq ReadyMix (Sigma-Aldrich), 5 µl of diluted cDNA, 0.2 µl of reference dye and 150 nM of each primer. Reaction mixtures were preheated at 95°C for 2 min followed by 40 cycles of melting at 95°C for 15 s, annealing at 60°C for 45 s, and elongation at 72°C for 45 s on a Mx3000P qPCR System (Agilent Technologies, Santa Clara, CA). The expression levels of targets genes were measured in technical triplicates and normalized to GAPDH (in birds [*Feng et al., 2010*]) or HPRT (in bats) gene expression. In bat TA and extraocular muscle tissues no reference gene (GAPDH, HPRT, tubulin) could be amplified and MYH expression in these tissues could thus not be normalized. The sequences of the primers used are provided in the *Supplementary file 1C, D*. Amplicons were sequenced (Eurofins Genomics, Germany) after all experiments to confirm identity. We only included genes where we had a positive control in the tissues sampled. Data for zebra finches were normalized to the maximum value of all muscles per gene, after which mean ± SE was

calculated. Because of the lower amount of muscle tissue present in females compared to males (*Wade and Buhlman, 2000*), we pooled the tissue of three females prior to tissue homogenization. To compare across juvenile male birds, we collected syringeal tissue of three individuals of the ages 25, 50, 75 and 100 DPH.

## TEM Volumetric percentage estimates (VPE)

The left syringeal muscle VTB and left TA was dissected in three adult male zebra finches. The left laryngeal muscle ACTM and left TA was dissected in two adult Daubenton's bats. Tissues were fixed with a 2.5% glutaraldehyde in 0.1 M sodium cacodylate buffer (pH 7.3) for 24 hr and subsequently rinsed four times in 0.1 M sodium cacodylate buffer. Following rinsing, muscles were post-fixed with 1% osmium tetroxide ($OsO_4$) and 1.5% potassium ferrocyanide ($K_4Fe(CN)_6$) in 0.1 M sodium cacodylate buffer for 90 min at 4°C. After post-fixation, muscles were rinsed twice in 0.1 M sodium cacodylate buffer at 4°C, dehydrated through a graded series of alcohol at 4–20°C, infiltrated with graded mixtures of propylene oxide and Epon at 20°C, and embedded in 100% Epon at 30°C. Ultra-thin (60 nm) sections were cut (Leica Ultracut UCT ultramicrotome, Leica Microsystems, Brønshoj, Denmark) at three depths (separated by 150 µm) and contrasted with uranyl acetate and lead citrate. Sections were examined and photographed at 40,000x magnification in a pre-calibrated transmission electron microscope (Philips EM 208 and a Megaview III FW camera or JEOL-1400 microscope (JEOL, Nieuw Vennep, The Netherlands) and a Quemesa camera (EMSIS GmbH, Münster, Germany)). We quantified volumetric percentage estimates (VPE) of sarcoplasmatic reticulum ($VPE_{SR}$), mitochondria ($VPE_{mt}$) and myofibrils ($VPE_{mf}$). The $VPE_{SR}$ and $VPE_{mito}$ and $VPE_{myo}$ were determined by point counting using grid sizes of 135, 135 and 300 nm, respectively (*Nielsen et al., 2010*). For zebra finch SFM the $VPE_{SR}$ and $VPE_{myo}$ were based on 168 images obtained from 14 fibres in 2 animals, and the $VPE_{mito}$ was based on 288 images obtained from 24 fibres in 3 animals (4,10,10). For zebra finch TA all the estimates were based on 61 images obtained from 5 fibres in 2 animals. For bat CT all the estimates were based on 80 images obtained from 7 fibres in 2 animals, and for bat TA all the estimates were based on 48 images from 4 fibres in 2 animals. The estimated coefficient of error for ratio estimators were 0.05, 0.03 and 0.08 for the estimates of SR, myofibrils and mitochondria, respectively. The rest volume consisted of myoplasm, lipids, nuclei and glycogen particles. SR per myofibril ratio was defined as $\frac{VPE_{SR}}{VPE_{mf}} * 100\%$. TEM images of the Toadfish white muscle body (as prepared in *Felder and Franzini-Armstrong, 2002*) for intraspecific comparison to toadfish swimbladder SFM were kindly provided by Dr. C. Franzini-Armstrong. Other values were taken from the literature: *Crotalus atrox* tailshaker and dorsal mid-body muscle (*Schaeffer et al., 1996*); and *Opsanus tau* swimbladder muscle (*Appelt et al., 1991*) and teleost white muscle (*Johnston, 1983*). Meaningful statistical comparison of the toadfish data could not be performed on published mean values in toadfish.

## Force measurements and real-time calcium imaging

Force development curves of zebra finch syringeal VTB muscle were measured at 50 (n = 4), 75 (n = 4) and 100 DPH (n = 4) at 39°C at optimal stimulus amplitude and muscle length as described previously (*Elemans et al., 2008*, *2011*). Real-time force development and intracellular calcium concentration kinetics were measured in four adult male zebra finches and two adult male toadfish on a custom-built microscope setup. Briefly, the muscle was suspended between a force transducer (model 400A, Aurora Scientific, Aurora, ON, Canada) and micromanipulator submerged in temperature controlled (accuracy ±0.1°C) Ringers solution. This muscle chamber was mounted on an inverted microscope (Nikon eTI, DFA, Glostrup, Denmark). First, force development curves of single and multiple stimuli (100 ms, 100 Hz trains) were measured at a range of temperatures (10–39°C) at optimal stimulus amplitude and muscle length. Stimulus trains were measured at 25°C and 39°C for toadfish and zebra finch respectively. Second, we flushed the sample with 20 µM BTS in Ringers for 20 min to inhibit actomyosin interaction and avoid movement artifacts for the subsequent calcium measurements (*Cheung et al., 2002*). Third, to image intracellular calcium, we used a high-affinity calcium dye (*Hollingworth et al., 2009*) (Mag-Fluo-4, AM stock, ThermoFisher scientific, Slangerup Denmark, # F14201) in DMSO, diluted to 2 µM in ringer's solution) and incubated the samples for 30 min at 21°C after which they were rinsed three times with indicator-free medium. A 20x air objective was used to image the sample using a Metal Halide lamp as illumination source and a FITC filter

cube (Nikon, DFA, Glostrup, Denmark) to isolate the calcium dye signal. We imaged a small portion of the sample to increase the signal to noise ratio. The resulting signal was passed through a 535 ± 15 nm filter (AHF, Germany) detected by a photo multiplier tube (Model R928, Hamamatsu, Denmark) and amplified (model SR560, Stanford Research Systems, Sunnyvale, CA). The initial temperate series was now repeated at identical stimulation settings. Force, stimulus, and calcium signals were digitized (National Instruments USB6259), aligned on the stimulus onset and analyzed in Matlab (RRID:SCR_001622). We performed these measurements in zebra finch syringeal VTB muscle (N = 5) and toadfish swimbladder muscle (N = 2). The bat laryngeal CT muscle's force data presented in *Figure 6b* was collected as part of a previous study (*Elemans et al., 2011*).

## SR vesicle Ca$^{2+}$ uptake and release

Because of limited tissue availability of bat muscle and the rapid experimental rundown of mammalian muscles *in vitro* at relevant physiological temperatures, we could not optimize the real-time calcium imaging technique for bats and used an alternative technique to measure free calcium uptake and release by isolated SR vesicles in muscle homogenate (*Nielsen et al., 2007*). Briefly, the SR vesicle oxalate mediated, calcium uptake and release was measured fluorometrically (Ratiomaster RCM, Photon Technology International, Brunswick, NJ) at 37°C using a fluorescent calcium indicator (indo-1). The [Ca$^{2+}$] release and uptake curves were fitted using exponential equations as previously described (*Nielsen et al., 2007*).

## Statistics

No sample sizes were computed before the experiments. A technical replicate is a replicate of the measurement on the same preparation, and a biological replicate is an individual. Data are presented as mean values ± 1 SE. *MYH13* mRNA levels in juvenile birds were compared using Kruskal-Wallis test, followed by Dunn's posthoc test. Juvenile twitch times were compared using Kruskal-Wallis test, followed by Wilcoxon signed rank tests. SR volume and SR per myofibril ratio were compared between each SFM and the intraspecific skeletal muscle using a Kruskal-Wallis test. Statistical test results for the rattlesnake TEM data were reported in *Schaeffer et al., 1996*. dF/Fr peak magnitude and FWHM values were compared between binned twitch number (1, 2–4 and 5–10) using a Kruskal-Wallis test. SR vesicle [Ca$^{2+}$] uptake and release data were compared between SFM and intraspecific reference muscle by one-way analysis of variance (ANOVA). The number of technical and biological replicates are listed above with each essay. No outliers or data were excluded.

## Acknowledgements

We thank T Christensen, P Martensen, K Lundgreen and S Jacobs for technical assistance, the MRC staff at the Marine Biological Laboratories, Woods Hole, and A Mensinger for supplying toadfish, A Surlykke, S Brinkløv, L Jakobsen for bat access, FH Andreade, C Moncman, N Rubinstein for discussion and MYH13 antibodies, C Franzini-Armstrong for toadfish body muscle TEM images, and S Baylor for calcium imaging advice. JM Ratcliffe, and RJ Schilder commented on the manuscript. This study was funded by grants from the National Institute of Arthritis and Musculoskeletal and Skin Diseases Training Grant T32 AR-053461 to AFM, Lundbeck Foundation to NO, NØ and BB, Danish National Research Council/Medical Sciences (FSS) to IK, Danish Research Council (FNU) and Carlsberg Foundation to CPHE.

## Additional information

### Funding

| Funder | Grant reference number | Author |
| --- | --- | --- |
| National Institute of Arthritis and Musculoskeletal and Skin Diseases | AR-053461 | Andrew F Mead |
| Lundbeckfonden | | Nerea Osinalde<br>Niels Ørtenblad<br>Blagoy Blagoev |

| | | |
|---|---|---|
| Sundhed og Sygdom, Det Frie Forskningsråd | | Irina Kratchmarova |
| Natur og Univers, Det Frie Forskningsråd | Sapere Aude 2 | Coen PH Elemans |
| Carlsbergfondet | | Coen PH Elemans |

The funders had no role in study design, data collection and interpretation, or the decision to submit the work for publication.

## Author contributions

Andrew F Mead, Conceptualization, Data curation, Formal analysis, Funding acquisition, Validation, Investigation, Visualization, Methodology, Writing—original draft, Writing—review and editing; Nerea Osinalde, Ulrik Frandsen, Data curation, Formal analysis, Investigation; Niels Ørtenblad, Data curation, Formal analysis, Supervision, Funding acquisition, Investigation, Writing—review and editing; Joachim Nielsen, Michiel Vellema, Iris Adam, Yafeng Song, Data curation, Formal analysis, Investigation, Writing—review and editing; Jonathan Brewer, Data curation, Formal analysis, Investigation, Methodology, Writing—review and editing; Constance Scharff, Blagoy Blagoev, Irina Kratchmarova, Resources, Supervision, Writing—review and editing; Coen PH Elemans, Conceptualization, Resources, Data curation, Software, Formal analysis, Supervision, Funding acquisition, Validation, Investigation, Visualization, Methodology, Writing—original draft, Project administration, Writing—review and editing

## Author ORCIDs

Andrew F Mead (iD) http://orcid.org/0000-0003-4716-2149
Joachim Nielsen (iD) http://orcid.org/0000-0003-1730-3094
Iris Adam (iD) http://orcid.org/0000-0001-7561-7636
Constance Scharff (iD) http://orcid.org/0000-0002-5792-076X
Ulrik Frandsen (iD) http://orcid.org/0000-0001-5816-4823
Blagoy Blagoev (iD) http://orcid.org/0000-0002-3596-0066
Irina Kratchmarova (iD) http://orcid.org/0000-0001-6613-7315
Coen PH Elemans (iD) http://orcid.org/0000-0001-6306-5715

## Ethics

Animal experimentation: All procedures were carried out in accordance with the Danish Animal Experiments Inspectorate (Copenhagen, Denmark), license #2013-15-2934-00991. All animals were euthanized humanely at the start of the protocol and did not experience any suffering or pain.

## Decision letter and Author response

Decision letter https://doi.org/10.7554/eLife.29425.013
Author response https://doi.org/10.7554/eLife.29425.014

# Additional files

## Supplementary files

• Supplementary file 1.
DOI: https://doi.org/10.7554/eLife.29425.011

• Transparent reporting form
DOI: https://doi.org/10.7554/eLife.29425.012

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
