## [Decision Letter]

Thank you for submitting your article "Fundamental Constraints In Synchronous Muscle Limit Superfast Motor Control In Vertebrates" for consideration by *eLife*. Your article has been reviewed by 3 peer reviewers, and the evaluation has been overseen by a Reviewing Editor and Eve Marder as the Senior Editor. One of the reviewers, Lawrence Rome, has agreed to share his identity.

The reviewers have discussed the reviews with one another and the Reviewing Editor has drafted this decision to help you prepare a revised submission.

Summary:

Superfast muscles (SFMs), the fastest known synchronous muscle of vertebrates, are known in fish, reptiles, birds and mammals where they are mainly linked to sound production. Early studies of SFMs that vibrate the toadfish swimbladder and rattlesnake tailshaker showed how the excitation-contraction coupling pathway underlying force generation in skeletal muscle allows an especially high rate of contraction and rapid modulation of sound pulses; the tradeoff being low force generation. These SFMs also exhibit a hypertrophy of the sarcoplasmic reticulum that allows for a more rapid recycling of calcium ions essential to the formation of actomyosin bridges that generate force, and thin myofibrils that reduce the diffusion distance for calcium. This new study characterizes the molecular basis for SFM physiological capacity and its evolutionary history by combining comparative genomic and quantitative (q) PCR analyses with morphometric and calcium dynamic analyses. To complement the prior studies of toadfish and rattlesnakes, this study includes the sound-producing SFMs of avian and mammalian species – the of zebra finch (ZF) syrinx and Daubenton's bat larynx. The results represent a novel contribution to our understanding of the functional architecture and evolutionary history of SFMs that are a prime example of innovations in skeletal muscle adapted to specific behavioral tasks.

Essential revisions:

1) It is sometimes unclear for the ZF studies if the dorsal or ventral tracheobronchialis muscle (DTB, VTB) is being studied: The text describing Figure 4 reports results for the VTB, but the figure is labeled DTB. DTB is also the label in Figure 4 but there is no specific reference to DTB or VTB in the text – which one is in Figure 4? Which one is shown in supplemental figure for Figure 4? Figure 5 refers to DTB – the text just refers to SFM. Figure 6 does not indicate if the results are for the VTB or DTB. The text and figures should be clear as to which muscle is used in each analysis. If the same muscle is not used throughout all of the analyses it should be explained why this is the case.

2) Discussion, first paragraph: The hypoglossal nerve is considered to be homologous to occipital nerve roots in fish (a hypoglossal is not identified in fish) so they are not as distinct as one might be led to think here – recent molecular evidence supports the hypothesis that sonic motor neurons in fish and birds arose from a common progenitor domain distinct from the one that gave rise to laryngeal motor neurons (Albersheim-Carter et al., 2016, Respiratory Physiology and Neurobiology 224: 2-10). This does not mean that the SFMS in birds and fish do not depend upon distinct MYH genes for their function and "descend from different evolutionary events" but the authors might want to be more circumspect in saying that occipital and hypoglossal nerves are "different cranial nerves".

3) The last paragraph of the Discussion should include a comment on the genomic analysis that supports independent origins for the MYH in each group studied.

4) There are too many abbreviations.

5) The authors should justify their selection of a 100Hz cycle frequency as the boundary for super-fast muscles.

---

## [Author Response]

1) It is sometimes unclear for the ZF studies if the dorsal or ventral tracheobronchialis muscle (DTB, VTB) is being studied: The text describing Figure 4 reports results for the VTB, but the figure is labeled DTB. DTB is also the label in Figure 4 but there is no specific reference to DTB or VTB in the text – which one is in Figure 4? Which one is shown in supplemental figure for Figure 4? Figure 5 refers to DTB – the text just refers to SFM. Figure 6 does not indicate if the results are for the VTB or DTB. The text and figures should be clear as to which muscle is used in each analysis. If the same muscle is not used throughout all of the analyses it should be explained why this is the case.

We apologize for these errors and thank the reviewers for catching them. In our experiments we measured MYH content using qPCR separately for all syringeal muscles in adult males and females. In this manuscript, we decided to only include data on the VTB, for which superfast muscle performance has been established earlier. Unfortunately, we erroneously presented the DTB qPCR data in Figure 4 and B. In our revision we have omitted DTB qPCR data and include only data on VTB. We rephrased the relevant Materials and methods section into:

“We dissected the syringeal muscle *tracheobronchialis ventralis* (VTB) on ice immediately after isoflurane euthanasia (Elemans et al., 2008).”

Consequently we replaced Figure 4 with only VTB data and reran the corresponding statistics presented in Figure 4 and main text. Importantly, none of the statistics changed and as such these changes have no effect on our conclusions.

Furthermore, in Figure 1, Figure 4—figure supplement 1, and Figure 5 we made an error in the labelling and have relabelled these figures. These measurements were all from VTB as mentioned correctly in the relevant methods section.

2) Discussion, first paragraph: The hypoglossal nerve is considered to be homologous to occipital nerve roots in fish (a hypoglossal is not identified in fish) so they are not as distinct as one might be led to think here – recent molecular evidence supports the hypothesis that sonic motor neurons in fish and birds arose from a common progenitor domain distinct from the one that gave rise to laryngeal motor neurons (Albersheim-Carter et al., 2016, Respiratory Physiology and Neurobiology 224: 2-10). This does not mean that the SFMS in birds and fish do not depend upon distinct MYH genes for their function and "descend from different evolutionary events" but the authors might want to be more circumspect in saying that occipital and hypoglossal nerves are "different cranial nerves".

We thank the reviewers for pointing out this very interesting paper. As this new evidence renders our previous statement incorrect, we decided to omit the sentence. We tried adding a brief statement regarding the origin and physiological properties of the vocal/sonic motorneurons to our Discussion both at the end of the first paragraph and also at the last paragraph before the concluding paragraph. However, in both cases this introduced such a significant break of the reading flow and detracted from the main point of the paragraph that we decided to not do so. In the revised manuscript, the first Discussion paragraph ends as follows:

“Since extremely fast crossbridge kinetics are necessary, though not sufficient, for superfast performance (Rome, 2006), and these are set largely by *MYH* gene expression in vertebrates (Bottinelli, Schiaffino and Reggiani, 1991), the parsimonious conclusion is that the SFM phenotype found in the sound producing organs of bats, songbirds, and toadfish descend from different evolutionary events.”

3) The last paragraph of the Discussion should include a comment on the genomic analysis that supports independent origins for the MYH in each group studied.

We have rephrased the concluding paragraph into:

“To conclude, the non-orthology of the dominant myosin heavy chains expressed in SFMs of birds, mammals, and fish indicates that SFM likely evolved independently in each lineage. […] Motor-driven acoustic modulation rates in complex communication are thus fundamentally limited by muscle architecture.”

4) There are too many abbreviations.

We acknowledge that we used many abbreviations and in our revision aimed to reduce the number of abbreviations as much as possible, especially in the abstract and main text. As such, we omitted six abbreviations in the main text (MY-32, EOM, EDL, VPE, TnC, VTB), but kept the five abbreviations relevant to this paper (ECC, MYH, qPCR, SFM, SR), and some that are widely used (ADP, ATP, Ca^2+^, SERCA -this abbreviation would otherwise be replaced by SR Ca^2+^-ATPase which is not helping readability). In the Discussion we kept essential gene and protein names (MyHC-sf, MyHC-eo, MyHC-2b).

5) The authors should justify their selection of a 100Hz cycle frequency as the boundary for super-fast muscles.

In our revised manuscript we use the definition by Rome (2006) where SFM range from approximately 90 to 250 Hz.

Introduction: “Their ability to repetitively contract and relax fast enough to produce work at cycling rates over around 90 Hz and up to 250 Hz sets them apart from other muscles by almost two orders of magnitude and allows the execution of central motor commands with millisecond temporal precision (Rome, 2006; Rome et al., 1988; Elemans et al., 2004; Elemans et al., 2008; Elemans et al., 2011.”

Figure 1 legend: “SFM (labeled purple) has been defined as synchronous muscle capable of producing work at cycling frequencies in excess of 90 Hz (grey box) (Rome, 2006).”